# Heteromers Formed by GPR55 and Either Cannabinoid CB_1_ or CB_2_ Receptors Are Upregulated in the Prefrontal Cortex of Multiple Sclerosis Patients

**DOI:** 10.3390/ijms25084176

**Published:** 2024-04-10

**Authors:** Carlota Menéndez-Pérez, Rafael Rivas-Santisteban, Eva del Valle, Jorge Tolivia, Ana Navarro, Rafael Franco, Eva Martínez-Pinilla

**Affiliations:** 1Department of Morphology and Cell Biology, University of Oviedo, 33006 Oviedo, Spain; menendezperezcarlota@gmail.com (C.M.-P.); valleeva@uniovi.es (E.d.V.); jtolivia@uniovi.es (J.T.); anavarro@uniovi.es (A.N.); 2Instituto de Neurociencias del Principado de Asturias (INEUROPA), 33006 Oviedo, Spain; 3Instituto de Investigación Sanitaria del Principado de Asturias (ISPA), 33006 Oviedo, Spain; 4Molecular Neurobiology Laboratory, Department of Biochemistry and Molecular Biomedicine, Faculty of Biology, University of Barcelona, 08028 Barcelona, Spain; rrivasbioq@gmail.com (R.R.-S.); rfranco123@gmail.com (R.F.); 5CiberNed, Network Center for Neurodegenerative Diseases, National Spanish Health Institute Carlos III, 28031 Madrid, Spain; 6Laboratory of Computational Medicine, Biostatistics Unit, Faculty of Medicine, Autonomous University of Barcelona, Campus Bellaterra, 08193 Bellaterra, Spain

**Keywords:** cannabinoids, endocannabinoid system, oligomerization, PLA, prefrontal cortex

## Abstract

Multiple sclerosis (MS) is an autoimmune, inflammatory, and neurodegenerative disease of the central nervous system for which there is no cure, making it necessary to search for new treatments. The endocannabinoid system (ECS) plays a very important neuromodulatory role in the CNS. In recent years, the formation of heteromers containing cannabinoid receptors and their up/downregulation in some neurodegenerative diseases have been demonstrated. Despite the beneficial effects shown by some phytocannabinoids in MS, the role of the ECS in its pathophysiology is unknown. The main objective of this work was to identify heteromers of cell surface proteins receptive to cannabinoids, namely GPR55, CB_1_ and CB_2_ receptors, in brain samples from control subjects and MS patients, as well as determining their cellular localization, using In Situ Proximity Ligation Assays and immunohistochemical techniques. For the first time, CB_1_R-GPR55 and CB_2_R-GPR55 heteromers are identified in the prefrontal cortex of the human brain, more in the grey than in the white matter. Remarkably, the number of CB_1_R-GPR55 and CB_2_R-GPR55 complexes was found to be increased in MS patient samples. The results obtained open a promising avenue of research on the use of these receptor complexes as potential therapeutic targets for the disease.

## 1. Introduction

Multiple sclerosis (MS) is a neurodegenerative, demyelinating, and inflammatory disease of the central nervous system (CNS) that affects more than 2.8 million people worldwide, with its prevalence three to four times higher in women than in men [1]. Its incidence is increasing; it is the second cause of disability among people between 20 and 30 years old, although it can occur in children and older adults [2]. Geographic area, as well as race and ethnicity, also appears to be related to the risk of developing the disease, with White people of European origin being the most affected [1]. MS is characterized by inflammation, progressive focal loss of oligodendrocytes, and demyelinating lesions known as plaques, in both the grey matter (GM) and white matter (WM), which compromise axonal transport and ultimately result in the typical symptomatology of the disease [3,4,5,6]. Symptoms include, but are not limited to, fatigue, tremors, spasticity, pain, bladder dysfunction, visual impairment, and cognitive deficits [7]. The heterogeneous clinical course delineates up to four MS types, i.e., clinically isolated syndrome (CIS), relapsing-remitting MS (RRMS), primary progressive MS (PPMS), and secondary progressive MS (SPMS). An early diagnosis of the initial episode is instrumental in prescribing individualized treatments and preventing multiple relapses [8]. To date, all attempts to identify a panel of biomarkers applicable to the disease have failed [3,9,10]. The exact cause of MS is unknown, but a variety of genetic and environmental factors, such as lifestyle and viral infections, may contribute to the onset of the disease [3,11]. While there is no cure for MS, translational research has provided effective treatments focused on reducing symptoms (e.g., glucocorticoids) or modifying the natural course of the disease (e.g., monoclonal antibodies and immunomodulatory agents) [12,13,14]. Despite symptomatic relief and, in some cases, the slowing of disease progression, all of these therapies fail in the long term and seriously compromise patients’ quality of life, both physically and cognitively. Cumulative clinical evidence has demonstrated that certain natural cannabinoids such as Δ^9^-tetrahydrocannabinol (THC) and cannabidiol (CBD) in a 1:1 mixture available and approved in the form of an oromucosal spray, Sativex^®^ (nabiximols), reduce MS-related spasticity [15]. Therefore, cannabinoids have aroused great interest since they may offer potential new treatment options for patients suffering from MS [16,17,18].

Endogenous cannabinoids acting through cannabinoid receptors, CB_1_ and/or CB_2_, are relevant neuromodulators. They participate in several events occurring in the CNS, from neural development to neurotransmission and synaptic plasticity [19,20,21]. The CB_1_ receptor (CB_1_R) is the most abundant G-protein-coupled receptor (GPCR) superfamily member in mammalian CNS, it is particularly abundant in the olfactory bulb, hippocampus, basal ganglia, and cerebellum [22,23,24]. Moderate/low levels of this receptor have been found in the cerebral cortex, amygdala, hypothalamus, some areas of the brainstem, and the spinal cord [22,25]. Regarding CB_2_ receptor (CB_2_R) expression in the CNS, it is primarily identified in microglia but also in some neurons in diverse brain areas including the cerebral cortex, hippocampus, and globus pallidus [23,26,27,28,29,30,31]. At the neuronal level, these two receptors are preferentially located in the plasma membrane of pre- and post-synaptic terminals, but they can appear in the soma and dendrites of glutamatergic and serotonergic neurons, as well as in GABAergic interneurons [25,29,32,33]. Cannabinoids, endogenous, natural, or synthetic, may limit excitotoxic damage and enhance synaptic plasticity via CB_1_R or may exert anti-inflammatory and immunomodulatory actions via CB_2_R [17,34]. Despite responding to cannabinoids, GPR55 (G protein-coupled receptor 55) is still considered an orphan receptor. Although early studies proved the binding of CP55940, a synthetic cannabinoid, to GPR55 and activation of this receptor by anandamide [35], it is now assumed that the binding of cannabinoids to the receptor occurs at an allosteric site [36,37,38]. GPR55 is abundant in various regions within the CNS, both in neurons and glial cells; it has been described in the hippocampus, thalamic nuclei, and basal ganglia of rodent, primate, and human brains [25,30,39,40]. While CB_1_R and CB_2_R couple primarily to the adenylyl cyclase-inhibiting heterotrimeric G_i/o_ protein [41,42,43], GPR55 likely couples to G_q/G11_ and G_12_/G_13_, showing a pleiotropic pharmacological profile that includes the activation of different Rho and ERK signal transduction pathways [41,44,45].

GPCRs are often expressed on the cell surface as dimers/oligomers, and the functionality of these receptor complexes, i.e., ligand binding and signaling characteristics, differ from when expressed as monomers [46,47,48]. Evidence of direct interactions between CB_1_R or CB_2_R and GPR55 has been obtained through the use of biophysical, biochemical, and pharmacological approaches. CB_1_R-GPR55 and CB_2_R-GPR55 receptor heteromers have been identified in in vitro models [49,50] and also in the rat striatum and in the caudate, putamen and accumbens nuclei of non-human primates [25,51]. Something that is noteworthy is that alterations in the brain levels of these heteromers have been demonstrated in animal models of Parkinson’s disease and in parkinsonian animals rendered dyskinetic by treatment with levodopa [30].

This study aimed to explore the possibility that CB_1_R-GPR55 and CB_2_R-GPR55 heteromers are expressed in the human prefrontal cortex. Taking into account that cannabinoids can ameliorate clinical signs of MS by exerting their effects through cannabinoid receptors [52,53], this study also determined whether the expression of these heteromers was altered in prefrontal cortex samples from patients affected by MS.

## 2. Results

### 2.1. Expression of CB_1_R-GPR55 Heteromers in Controls and MS Patients

The In Situ Proximity Ligation Assay (PLA) is the most appropriate technique to detect the presence of complexes formed by two different receptors in a native system. Using specific antibodies against the CB_1_R and the GPR55 coupled to PLA oligonucleotide probes, a punctate fluorescent red signal was observed surrounding DAPI-counterstained nuclei, reflecting the formation of CB_1_R-GPR55 heteromers in the plasma membrane of cells in all prefrontal cortex samples analyzed (see Appendix A), both in GM and WM (Figure 1A). Data analysis demonstrated that the number of red dots per cell, which reflects the number of CB_1_R-GPR55 heteromers, was significantly higher in the GM compared to the WM in the prefrontal cortex samples from the controls and MS patients (F_1,596_ = 8.23; *p* < 0.05) (Figure 1B).

Once the expression of CB_1_R-GPR55 heteromers in the human prefrontal cortex was shown, the next objective was to determine whether their expression was altered in samples from MS patients. The representative images in Figure 1A demonstrate an increase in the amount of red signal and, consequently, the expression of CB_1_R-GPR55 heteromers in samples from the MS patients (GM and WM). Quantitative analysis confirmed a significantly higher density of red clusters in the prefrontal cortex of MS patients compared to data obtained using samples from healthy controls (Figure 1B).

### 2.2. Expression of CB_2_R-GPR55 Heteromers in Controls and MS Patients

Similar findings were found when analogous assays addressing CB_2_R-GPR55 heteromer formation were performed using the same brain samples. Representative images and quantitative data from PLA assays (Figure 2A) revealed that CB_2_R and GPR55 formed heteromeric complexes in the plasma membrane of cells (see Appendix A) in all brain areas and samples analyzed. When considering the total number of CB_2_R-GPR55 heteromers by assessing the number of red dots per cell, data analysis demonstrated higher expression in the GM of the prefrontal cortex compared to the WM, both in the control and MS patient samples (F_1,572_ = 4.63; *p* < 0.05) (Figure 2B).

Once again, significant differences were found when comparing the red fluorescent dots in the controls and individuals diagnosed with MS; the prefrontal cortex of patients showed a greater amount of CB_2_R-GPR55 heteromers compared to that observed in the controls (Figure 2A,B).

### 2.3. Expression of CB_1_R-GPR55 and CB_2_R-GPR55 Heteromers in Neurons and Different Types of Glial Cells

The final experimental approach was designed to identify those cells in the human prefrontal cortex that expressed CB_1_R-GPR55 or CB_2_R-GPR55 complexes. While the PLA was used to detect the presence of receptor–receptor interactions, neurons and glial cells were visualized using classical immunohistochemical staining. The combination of the two approaches made it possible to detect CB_1_R-GPR55 heteromers in a significant number of NF-200-labeled neurons in the samples of control individuals (Figure 3A–C). However, no co-staining was found when glial marker signals (GFAP, αβ-crystallin, and iba-1) were used (Figure 3D–L).

Regarding CB_2_R-GPR55 heteromers, the results were similar, i.e., the receptor complexes in the prefrontal cortex of the control subjects were identified in neurons (Figure 4A–C) but not in astrocytes, oligodendrocytes or microglia (Figure 4D–L).

## 3. Discussion

Understanding the modulatory role of the ECS in the CNS and its implication in neurodegenerative diseases has been the focus of considerable research efforts over the past few decades [19]. Despite the benefits of cannabinoids and the approval of cannabinoid-based medication, Sativex^®^, for some symptoms, little is known about the role of the ECS in the etiology and progression of MS. Recent evidence using animal models of this pathology sustained the therapeutic potential of cannabinoids in reducing certain symptoms of MS through the activation of cannabinoid receptors [54]. Preclinical studies in the experimental autoimmune encephalomyelitis (EAE) mouse model demonstrated that treatment with CBD and THC decreases axonal damage, inflammation, microglial activation, and T-cell recruitment, leading to a symptomatic improvement that seems to be related to direct action on the CB_1_R [55,56]. Spasticity increases rapidly after the administration of a CB_1_R antagonist, rimonabant, suggesting that alleviation of hind limb alteration is CB_1_R dependent [57]. A more recent study reported in a cuprizone-induced mouse model of MS that reducing the global amount of CB_1_R limits myelin repair potential [52]. Regarding CB_2_R, some studies have shown that JWH-133, a selective receptor agonist, ameliorates tremor and spasticity in EAE mice by promoting autophagy and inhibiting NLRP3 inflammasome activation [53]. Similarly, treatment with another CB_2_R agonist, HU-308, attenuated the development of the pathological condition. Consistent with these observations, CB_2_R^−/−^ mice displayed greater vulnerability to neurofilament degeneration, inflammation, apoptosis, and axonal damage, common pathological features of the EAE [54]. Moreover, it is increasingly recognized that CBD and THC, administered together in controlled doses (Sativex^®^), reduce muscle spasms and spasticity in MS patients and even induce analgesia [15]. However, the pharmacology behind the receptor-mediated neuroprotective effects exerted by cannabinoids is not straightforward; it involves multiple targets and mechanisms that, collectively, unfold a particular pattern of cellular events. In this sense, dimer/oligomerization is now considered a relevant mechanism to induce diverse functional selectivity in signaling mediated by GPCRs [47]. Considering the ability of cannabinoid receptors to form heteromers that may constitute therapeutic targets, as already postulated for neurodegenerative diseases such as Alzheimer’s disease and Parkinson’s disease [30,58], this work aimed to adequately characterize the formation of complexes between CB_1_ or CB_2_ and GPR55 receptors in the CNS and to evaluate whether the expression of these receptor complexes is affected in MS.

The results presented herein constitute the first description of CB_1_R-GPR55 and CB_2_R-GPR55 heteromers in the human prefrontal cortex of control individuals and patients with MS. By taking advantage of the PLA technique and immunohistochemistry using antibodies against neuronal and glial markers, the expression of these receptor complexes was confirmed in neurons of the cerebral cortex, both in GM and WM, but not in glial cells labeled with antibodies against αβ-crystallin, GFAP or Iba-1. The possibility of direct interaction between CB_1_ or CB_2_ and GPR55 receptors has previously been demonstrated in cell cultures using energy transfer techniques [49,50] and cell biology techniques using brain samples of rats and non-human primates [25,30,51]. Indeed, CB_1_R-GPR55 heteromers were identified on the cell surface and in intracellular locations of striatal neuronal subtypes [25]. The expression of CB_2_R in the neurons of the CNS has been less well characterized, and the described changes in CB_2_R-GPR55 heteromer levels in the striatum of the *Macaca fascicularis* model of Parkinson’s disease were attributed to the upregulation of heteromers in activated microglia [30]. Interestingly, our data show that CB_1_R-GPR55 and CB_2_R-GPR55 heteromers are expressed almost exclusively at the level of the neuronal plasma membrane in the prefrontal cortex. Of note, the presence of these complexes is scarce in neuronal extensions which would explain the higher amount of heteromers found in GM compared to WM.

Remarkably, we identified an increase in the single-cell density of receptor complexes, for both CB_1_R-GPR55 and CB_2_R-GPR55, in the prefrontal cortex of MS patients (compared to control subjects). Changes in the expression of cannabinoid heteroreceptor complexes have been demonstrated in the brain of animal models of other neurodegenerative diseases [30,50]. An increase in the expression levels of CB_1_R-GPR55 and CB_2_R-GPR55 heteromers was found in basal ganglia input nuclei (i.e., caudate, putamen, and accumbens) of MPTP-treated parkinsonian primates; this increase was reverted through chronic treatment with levodopa only in those animals that became dyskinetic due to the chronic treatment [30]. The CB_1_R-CB_2_R heteromer is upregulated in activated microglia that, unlike resting microglia, are highly responsive to cannabinoids [58]. Interestingly, CB_1_R-CB_2_R heteromers were also upregulated in the hippocampus of a transgenic model of Alzheimer’s disease; it has been suggested that microglia in these animals display a neuroprotective phenotype that could explain why cognitive deficits do not appear until late in the life of the transgenic Alzheimer’s disease model [58]. The relevance of increased heteromer appearance to the pathophysiology of MS is unclear but could be part of a compensatory mechanism to restore homeostasis and brain integrity in response to neuronal damage. Cannabinoids are important players in neuroinflammation by regulating the release of neuropeptides and the activation of microglia [59,60]. In addition, these compounds may affect cellular energy production via GPR55-containing receptor complexes; mitochondrial dysfunctions observed in the disease could be caused by functional changes derived from differential heteromer expression [50,61]. All of this evidence supports the neuroprotective effect attributed to cannabinoids acting on cannabinoid and GPR55 receptors [50]. In this context, endocannabinoids and natural/synthetic cannabinoids capable of activating CB_1_R, CB_2_R, and GPR55 appear to offer protection against excitotoxic damage [17,38]. Furthermore, some studies in murine models have described a significant increase in endocannabinoid levels i.e., anandamide, palmitoylethanolamide, and 2-arachidonoylglycerol, as a part of an anti-inflammatory response resulting from axonal damage [19]. The presence of heteromers at significant levels in MS also opens up new possibilities in drug discovery, that is, targeting them for therapeutic benefit. The study has a limitation which is the small sample size derived from the difficulty in obtaining samples from patients. Although it may take time, validating MS-related changes in heteromer expression requires research with larger human cohort samples.

The potential of cannabinoids as drugs to combat or delay the progression of a neurological disease such as MS has gained interest in recent years. In the field of Parkinson’s disease research, it is increasingly recognized that targeting neuronal CB_1_R-GPR55 and CB_2_R-GPR55 heteromers with cannabinoids can be a successful therapeutic approach to both manage symptoms and delay disease progression [62,63]. The beneficial effect of some phytocannabinoids on MS symptoms [15] may be due to multiple molecular mechanisms. Cannabinoids can not only drive individualized responses through CB_1_, CB_2_, or GPR55 receptors but also, as this work suggests for the first time, act on functional units consisting of receptor heteromers. What is crucial to designing effective approaches is to consider the particular properties of the heteromers in terms of signaling. GPR55-mediated signaling is complex and this issue is delaying the discovery of selective compounds and the development of drugs targeting it. In contrast, the sustained interest in CB_2_R as a therapeutic target for neuroprotection has gained momentum over the last decade; agonists, antagonists, and allosteric modulators have been designed that, unlike CB_1_R targeting, lack unwanted psychotropic side effects [17,64]. Also interesting for the design of therapies to combat MS is the finding that CB_1_R-GPR55 and CB_2_R-GPR55 heteromers are expressed in neurons. This piece of information related to the CB_2_R-GPR55 heteromer is both intriguing and relevant since it is considered that the CB_2_R in the CNS is expressed more in the glia than in neurons. Finally, it should be noted that the existence of differences in the number of heteromers when comparing samples from control individuals and patients with MS offers a way forward for future research. In this sense, correlating changes in CB_1_R/CB_2_R-GPR55 heteromer levels with specific MS symptoms holds promise in using these complex receptors as therapeutic targets for personalized medicine approaches.

## 4. Materials and Methods

### 4.1. Subjects

In the present study, human prefrontal cortices from healthy subjects and patients with MS were used. It should be noted that brain samples from MS patients are very scarce. Human brain tissues were provided by different Spanish brain banks located at the University Hospital of Asturias, Central Nervous Tissue Bank Madrid (CIEN Foundation), the Center for Biomedical Research of Navarra (NAVARRABIOMED), and the Southern Galicia Health Research Institute (IISGS). In total, samples from eight individuals between 38 and 66 years old were obtained, including controls and those with histologically inflammatory demyelination consistent with MS, properly confirmed by a neuropathologist. Detailed information on the subjects and samples is given in Table 1.

Following retrieval, the brain specimens were fixed by immersion in 10% buffered formalin, dehydrated, cleared in butyl acetate, and embedded in paraffin. Tissue blocks containing the prefrontal cortex were sectioned at 7 µm, mounted on SuperFrost Plus (Mentzel-Glasse) slides, dried at 36 °C, and stored at room temperature until processed. 

The ethics committees of each participating biobank reviewed and approved the study protocol. Moreover, all research procedures involving the manipulation of human samples were approved by the *Comité de Ética de la Investigación del Principado de Asturias* (*CEImPA 23-174*) and are in accordance with the ethical principles and guidelines established in the Declaration of Helsinki and in the Spanish laws: *Ley de Investigaciones Biomédicas* (*ley 14/2007*) and *Ley de Protección de Datos Personales y Garantías de los Derechos Digitales* (*Ley Orgánica 3/2018*).

### 4.2. In Situ Proximity Ligation Assay (PLA)

In Situ PLA, a technique instrumental for detecting receptor–receptor interactions and their precise anatomical localization, was used to test for the presence of CB_1_R-GPR55 and CB_2_R-GPR55 heteromers in the prefrontal cortex of brain samples from control individuals and patients diagnosed with MS. For this purpose, the tissue sections were incubated for 1 h at 37 °C with the blocking solution, followed by overnight incubation at 4 °C with the PLA probe-linked antibodies (at a final concentration of 75 µg/mL). Proximity probes consist of affinity-purified antibodies modified by covalent attachment of the 5′ end of various nucleotides to each primary antibody. In this case, PLA probes were prepared by conjugating a rabbit anti-CB_1_ antibody (PA1-743, Invitrogen, Paisley, UK) and a rabbit anti-CB_2_ antibody (101550, Cayman Chemical, Ann Arbor, MI, USA) with a PLUS oligonucleotide (Duolink^®^ In Situ Probemaker PLUS ref: DUO92009; Sigma-Aldrich, St. Louis, MO, USA) and a rabbit anti-GPR55 antibody (10224; Cayman Chemicals, Ann Arbor, MI, USA), raised against the human 207–219 sequence, with a MINUS oligonucleotide (Duolink^®^ In Situ Probemaker MINUS Ref: DUO92010; Sigma-Aldrich, St. Louis, MO, USA) according to the manufacturer’s guidelines. After washing with buffer A (DUO82047; Sigma-Aldrich, St. Louis, MO, USA), the GPCR heteromers were detected using the Duolink in Situ PLA detection kit (Duolink^®^ In Situ Detection Reagents Red; DUO92008, Sigma-Aldrich, St. Louis, MO, USA). Then, the sections were washed with buffer A and incubated with the ligation solution for 1 h at 37 °C in a humidity chamber, washed with buffer A again, incubated with the amplification solution for 100 min at 37 °C, and finally washed with buffer B (wash buffer B; DUO82048; Sigma-Aldrich, St. Louis, MO, USA). The sections were finally mounted using an aqueous mounting medium with DAPI which allows for visualization of the cell nuclei (NB-23-00159-2, NeoBioTech, Nanterre, France). Appropriate negative control assays were carried out to ensure that there was a lack of non-specific labeling and amplification.

The quantification of PLA signals and cell nuclei was performed in images generated from a Leica sTCS-SP8X Spectral Confocal Laser Microscope (Leica Microsystems, Mannheim, Germany). Regarding selected regions of interest (ROIs), and for each field of view, a stack of two channels (one per staining) and 9–15 Z stacks with a step size of 1 μm were acquired with the 63× oil-immersion lens. Statistical analysis on the receptor heteromer densities was conducted according to a modification of the method of Tolivia et al. (see [65]). A quantification of cells containing one or more red spots versus total cells (blue nucleus) and the ratio r (number of red spots/cell) in cells containing spots were determined considering a total of 300–400 cells from ten different fields in both WM and GM for each prefrontal cortex section. The experiments were performed on a blind basis; the experimenter was not aware of the label and the conditions (control or MS) when images were taken. Moreover, the experimenter who made the analysis did not know the exact nature of the analyzed samples.

### 4.3. Co-Staining Combining PLA and Immunohistochemistry

Identification of the specific cell types (neurons, oligodendrocytes, astrocytes, and microglia) expressing CB_1_R-GPR55 and CB_2_R-GPR55 heteromers was carried out using immunohistochemistry followed by PLA. First, chromogenic immunodetection was performed using specific neuronal and glial markers. The immunohistochemistry process was conducted as follows. Prefrontal cortex sections of the control subjects were sequentially treated with Triton X-100 (0.01%, 5 min), washed with distilled water, treated with H_2_O_2_ (3%, 5 min), washed with distilled water again, and treated with PBS (2 min). Non-specific binding was blocked via incubation with 1% BSA (30 min). Then, incubation with a specific monoclonal antibody against NF-200, a neuronal marker (1:100; N-0142, Sigma-Aldrich, St. Louis, MO, USA), rabbit antibody against the glial fibrillary acidic protein (GFAP), an astrocytic marker (1:500; z0334, DAKO Agilent, Santa Clara, CA, USA), goat antibody against Iba-1, a microglia marker (1:1000; ab107159, Abcam, Cambridge, UK), and rabbit antibody against αβ-crystallin, a mature OLG marker (1:200; NCL-ABCrys-512, Novocastra, St. Gallen, Switzerland), was carried out overnight at 4 °C. After several washes in PBS, the sections were incubated at room temperature using a biotinylated horse universal antibody (1:40; PK-8800, Vector Laboratories Inc., Newark, NJ, USA) for 30 min. Afterward, the sections were treated with Extravidin labeled with horseradish peroxidase (HRP) (E2886, Sigma-Aldrich Extra-3, Sigma-Aldrich, St. Louis, MO, USA). Peroxidase activity was visualized with diaminobenzidine (DAB) (D4168, Sigma-Aldrich, St. Louis, MO, USA). After the immunohistochemical protocol, the PLA technique was used, as described in Section 4.2, to detect CB_1_R-GPR55 and CB_2_R-GPR55 heteromers.

The sections were observed using an Epi-Fl Nikon Eclipse E400 microscope (Nikon, Minato-ku, Tokyo, Japan) equipped with Plan Fluor objectives, and images were recorded using a digital camera (63×; NikonDN-100, Nikon, Minato-ku, Tokyo, Japan). Final images were obtained through the digital superposition of the corresponding DAB (NF-200, GFAP, αβ-crystallin, or Iba-1 signal) and red fluorescence (PLA signal) images of the same sections. The positive signal of each image was selected according to the method of Navarro et al. (2008) [66]; DAB signals were converted to green and saved as an RGB image. Merged images show the PLA fluorescence signal in red and the DAB label in green; the yellow color indicates the superposition of red and green colors.

### 4.4. Data Analysis

Data collected in samples from 4 subjects per group (control and MS) were the mean ± SEM. Statistical analysis was performed with GraphPad Prism 8 (San Diego, CA, USA). A two-way ANOVA followed by Bonferroni’s post hoc multiple comparison test were used to compare the values (r spots/cell) obtained for each pair of receptors. Data were tested for normality of populations and homogeneity of variances. Differences were considered significant when *p* < 0.05.

## Figures and Tables

**Figure 1 ijms-25-04176-f001:**
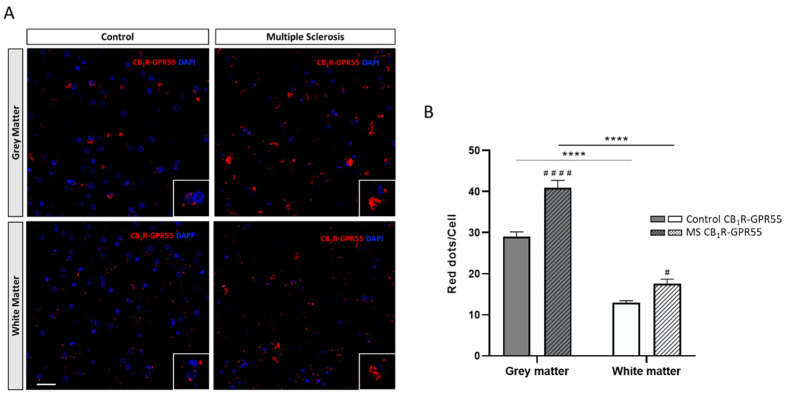
CB_1_R-GPR55 heteromers detected using the In Situ Proximity Ligation Assay (PLA) in the grey (GM) and white matter (WM) of the prefrontal cortex of samples from the controls and multiple sclerosis (MS) patients. Representative confocal images (40×) showing PLA label for CB_1_R-GPR55 heteromers as red dots in cells with DAPI-stained nuclei. Scale bar 50 µm (**A**). Quantification of CB_1_R-GPR55 heteromers as the number of red dots per cell in the GM and WM in the prefrontal cortex samples of the controls and MS patients. Data are the mean ± SEM (40 fields per section) (**B**). Significant differences were analyzed via two-way ANOVA followed by post hoc Bonferroni’s test. **** *p* < 0.0001 compared with GM; ^#^
*p* < 0.05, ^####^
*p* < 0.0001 compared with the control.

**Figure 2 ijms-25-04176-f002:**
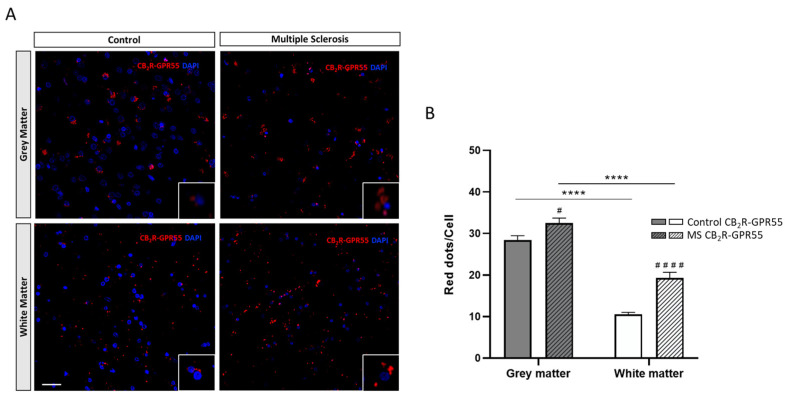
CB_2_R-GPR55 heteromers detected using the In Situ Proximity Ligation Assay (PLA) in the grey (GM) and white matter (WM) of the prefrontal cortex of samples from the controls and multiple sclerosis (MS) patients. Representative confocal images (40×) showing the PLA label for CB_2_R-GPR55 heteromers as red dots in cells with DAPI-stained nuclei. Scale bar 50 µm (**A**). Quantification of CB_2_R-GPR55 heteromers as the number of red dots per cell in the GM and WM in the prefrontal cortex samples of the controls and MS patients. Data are the mean ± SEM (40 fields per section) (**B**). Significant differences were analyzed via two-way ANOVA followed by post hoc Bonferroni’s test. **** *p* < 0.0001 compared with GM; ^#^
*p* < 0.05, ^####^
*p* < 0.0001 compared with the control.

**Figure 3 ijms-25-04176-f003:**
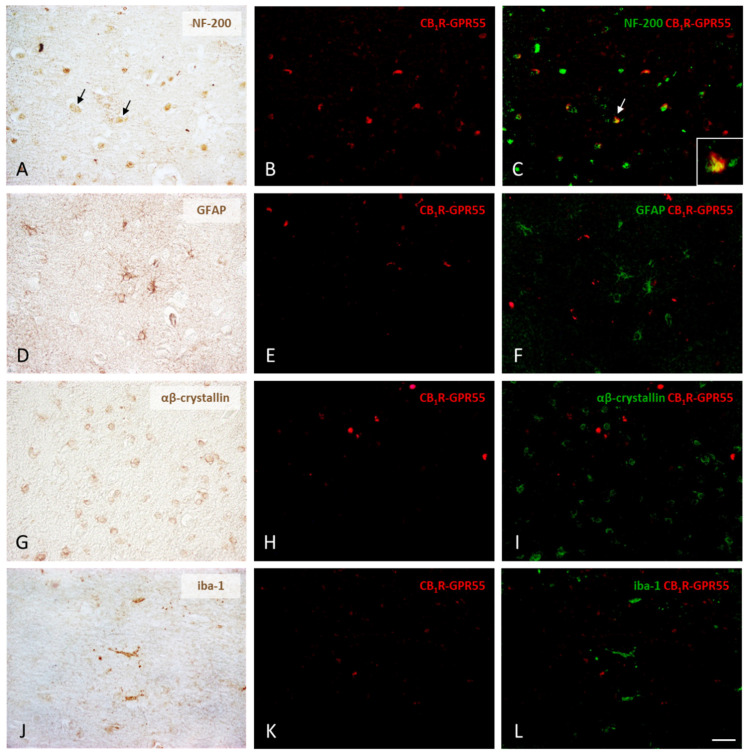
Expression of CB_1_R-GPR55 heteromers in neurons, astrocytes, oligodendrocytes, and microglia in the prefrontal cortex of the control subjects. Following the chromogenic immunohistochemical detection of the four different types of cells, those positive for NF-200, GFAP, αβ-crystallin, and iba-1 (**A**,**D**,**G**,**J**), CB_1_R-GPR55 heteromers were identified as red dots in cells with DAPI-stained nuclei using the PLA method (**B**,**E**,**H**,**K**). Digital overlays of PLA images (red fluorescent signal) and DAB immunolabeling images (the DAB signal converted into a green color) clearly show that only neurons express CB_1_R-GPR55 receptor complexes (arrows), whereas oligodendrocytes, astrocytes, and microglia completely lack them (**C**,**F**,**I**,**L**). 40×. Detail: 100×. Scale bar 50 µm.

**Figure 4 ijms-25-04176-f004:**
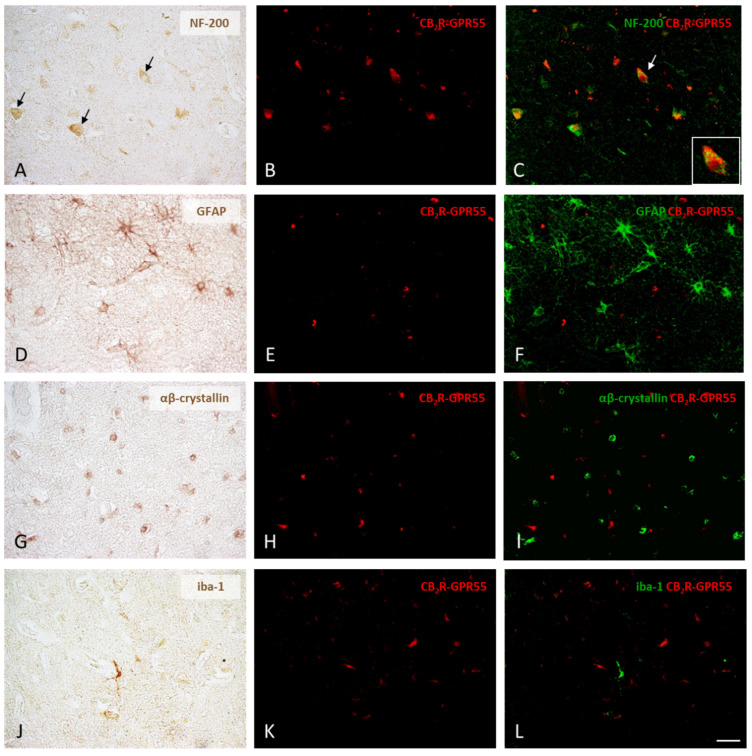
Expression of CB_2_R-GPR55 heteromers in neurons, oligodendrocytes, astrocytes, and microglia in the prefrontal cortex of the control subjects. Following the chromogenic immunohistochemical detection of the four different types of cells, those positive for NF-200, GFAP, αβ-crystallin, and iba-1 (**A**,**D**,**G**,**J**), CB_2_R-GPR55 heteromers were identified as red dots in cells with DAPI-stained nuclei using the PLA method (**B**,**E**,**H**,**K**). Digital overlays of PLA images (red fluorescent signal) and DAB immunolabeling images (the DAB signal converted into a green fluorescent signal) clearly showed that only neurons express CB_2_R-GPR55 receptor complexes (arrows), whereas oligodendrocytes, astrocytes, and microglia completely lack them (**C**,**F**,**I**,**L**). 40×. Detail: 100×. Scale bar 50 µm.

**Table 1 ijms-25-04176-t001:** Demographic data of cases.

Case	Sex	Age	Type of MS	Brain Area	Post-Mortem Time	Brain Bank
1	Male	58	MS	Prefrontal cortex	6–12 h	Central Nervous Tissue Bank Madrid (CIEN Foundation)
2	Male	38	SPMS	Prefrontal cortex	6–12 h	University Central Hospital of Asturias (HUCA)
3	Female	47	PPMS	Prefrontal cortex	6–12 h	Center for Biomedical Research of Navarra (NAVARRABIOMED)
4	Female	48	MS	Prefrontal cortex	6–12 h	Southern Galicia Health Research Institute (IISGS)
5	Male	38	Control	Prefrontal cortex	6–12 h	University Central Hospital of Asturias (HUCA)
6	Male	66	Control	Prefrontal cortex	6–12 h	Center for Biomedical Research of Navarra (NAVARRABIOMED)
7	Male	62	Control	Prefrontal cortex	6–12 h	Center for Biomedical Research of Navarra (NAVARRABIOMED)
8	Female	52	Control	Prefrontal cortex	6–12 h	University Central Hospital of Asturias (HUCA)

MS, multiple sclerosis; PPMS, primary progressive MS; SPMS, secondary progressive MS.

## Data Availability

The datasets generated and/or analyzed during the current study are not publicly available due to privacy/ethical restrictions but are available from the corresponding author upon reasonable request.

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
