# Peer review of "Heteromers Formed by GPR55 and Either Cannabinoid CB1 or CB2 Receptors Are Upregulated in the Prefrontal Cortex of Multiple Sclerosis Patients"

_ijms, 2024, doi:10.3390/ijms25084176_

Round 1

Reviewer 1 Report

Comments and Suggestions for Authors

In the paper by Carlota Menéndez-Pérez et al., titled “Characterization of CB1R-GPR55 and CB2R-GPR55 Heteromers in the Human Prefrontal Cortex: Implications for Multiple Sclerosis” the authors investigated the presence and alteration of CB1R-GPR55 and CB2R-GPR55 heteromers in the prefrontal cortex of individuals with multiple sclerosis. Using human brain samples from control subjects and MS patients, the study employs In Situ PLA and immunohistochemistry techniques. The results suggest an increase in the expression of these heteromers in the prefrontal cortex of MS patients compared to controls. The findings hold promise for understanding the role of endocannabinoid signaling in MS pathology and offer potential therapeutic targets for the disease. The manuscript investigates a relatively understudied area of cannabinoid receptor interactions in the context of MS, providing novel insights into potential therapeutic targets for the disease.

However, several issues need to be addressed:

Major issues:

1.      Introduction: please provide more detailed demographic and clinical characteristics of the MS patients to better contextualize the findings.

2.      Methods and results: While the study acknowledges the scarcity of human brain samples from MS patients, the small sample size might limit the generalizability of the findings. Future studies with larger cohorts or in animal models could further validate the observed alterations in heteromer expression. Please make a section limitation of the study or expand the sample size to be more relevant.

3.      The manuscript focuses primarily on the expression patterns of CB1R-GPR55 and CB2R-GPR55 heteromers in MS patients without delving into their functional implications. It focuses only on expression and carries only a description of the expression phenomenon on neurons. Including functional assays or correlating the heteromer expression with clinical parameters could enhance the mechanistic understanding.

4.      Discussion: Expand on the potential mechanisms underlying the observed alterations in heteromer expression, their relevance to MS pathophysiology, and the therapeutic implications, providing a more nuanced discussion.

Minor issues:

The authors should provide information on how the co-staining was assessed.

Line 102, please decipher the abbreviation PLA since it appears here for the first time.

Check all abbreviations.

Little things to be fixed.

Author Response

Point by point response to reviewers' comments:

Reviewer 1

Major issues:

1. Introduction: please provide more detailed demographic and clinical characteristics of the MS patients to better contextualize the findings.

Response: thanks for raising this issue that has been considered (briefly and with new references) in the revised version of the manuscript.

2. Methods and results: While the study acknowledges the scarcity of human brain samples from MS patients, the small sample size might limit the generalizability of the findings. Future studies with larger cohorts or in animal models could further validate the observed alterations in heteromer expression. Please make a section limitation of the study or expand the sample size to be more relevant.

Response: we appreciate the reviewer's understanding of the difficulty in obtaining samples of human brain tissue from patients with MS, and we agree that further studies in larger cohorts are needed to definitively validate the results. This issue has been addressed in the revised version of the manuscript according to the reviewer's suggestion (limitations). In this sense, we would like to mention that, once the formation of CB1R-GPR55 and CB2R-GPR55 heteromers in the human prefrontal cortex and the changes they undergo with the disease have been demonstrated, we are currently trying to increase the number of samples from MS patients to conduct a more comprehensive study that distinguishes between genders and different types of MS.

3. The manuscript focuses primarily on the expression patterns of CB1R-GPR55 and CB2R-GPR55 heteromers in MS patients without delving into their functional implications. It focuses only on expression and carries only a description of the expression phenomenon on neurons. Including functional assays or correlating the heteromer expression with clinical parameters could enhance the mechanistic understanding.

Response: thank you for the comment. Unfortunately, analyzing the functional implications of heteromerization in humans is a technical challenge. However, the functional consequences of the formation of these heteromers have already been widely described in in vivo and in vitro models. Correlating the expression of heteromers with human clinical parameters is certainly interesting, although it is beyond the scope of this work. In this regard, as mentioned in the previous query, we are currently conducting a more comprehensive and translational study that distinguishes between genders and different types of MS but also relates the expression of heteromers to MS symptoms such as spasticity and cognitive deficits.

4. Discussion: Expand on the potential mechanisms underlying the observed alterations in heteromer expression, their relevance to MS pathophysiology, and the therapeutic implications, providing a more nuanced discussion.

Response: we appreciate the suggestion that has been taken into account and we have added this information to the revised version of the manuscript. This is also related to the previous query.

Minor issues:

The authors should provide information on how the co-staining was assessed.

Response: thanks for the comment. We have modified the title of section 4.3 (in Methods) and rewritten the beginning of the section in the revised version of the manuscript. We think that the co-staining is now better explained. It consisted of first immunohistochemistry (using antibodies against different neuron/glial markers) followed by PLA (detailed in the previous section, 4.2).

Line 102, please decipher the abbreviation PLA since it appears here for the first time.

Response: thanks for detect this error that has been fixed in the revised version of the manuscript.

Check all abbreviations. Little things to be fixed.

Observation: major changes are highlighted in yellow. Grammatical corrections are not highlighted.

Reviewer 2 Report

Comments and Suggestions for Authors

The manuscript entitled “Heteromers Formed by GPR55 and either Cannabinoid CB1 or  CB2 Receptors Are Upregulated in the Prefrontal Cortex of Multiple Sclerosis Patients” describes staining for heteromers in the post-mortem brains of controls and MS individuals. Major strengths of the paper are the ability to make the assessment in human brains, the use of a novel staining technique to detect heterodimers, and the application of this staining technique to identify GPR55 heterodimerization with either CB1 or CB2. Some issues should be addressed.

I could not access the supplementary videos.

Is the DAPI signal also decreased in the MS patients? Possible reasons for this should be discussed. It seems like it might have been more difficult to quantify the red dots per cell in MS patients if DAPI was the definition of a cell (although I now see in M and M that it was total blue cells, not that a red signal had to necessarily be localized to a blue signal). Perhaps this can also be discussed, including the possibility that it might be this one individual in which this was the case.

In M and M, it is not clear what ab-crystallin stain defines.

Minor

Line 60, “The” should be “They”; also is it true they participate in every event? Consider using “many” or “most”

Line 74, “is” should be “being”

Line 121, “proved” should be “shown”

Line 122, “is” should be “was”; tenses are mixed throughout; consider changing all descriptions of results to past tense

Line 198, “consistenly” should be “consistent”

Comments on the Quality of English Language

minor edits as outlined 

Author Response

Point by point response to reviewers' comments:

The manuscript entitled “Heteromers Formed by GPR55 and either Cannabinoid CB1 or CB2 Receptors Are Upregulated in the Prefrontal Cortex of Multiple Sclerosis Patients” describes staining for heteromers in the post-mortem brains of controls and MS individuals. Major strengths of the paper are the ability to make the assessment in human brains, the use of a novel staining technique to detect heterodimers, and the application of this staining technique to identify GPR55 heteromerization with either CB1 or CB2. Some issues should be addressed.

 Response: we appreciate the positive comments.

I could not access the supplementary videos.

 Response: sorry for this inconvenience, we have contacted the IJMS Editorial Office and they have responded that the supplementary videos are valid and accessible. They suggest that the issue might be with the software the reviewer is using to access them.

 Is the DAPI signal also decreased in the MS patients? Possible reasons for this should be discussed. It seems like it might have been more difficult to quantify the red dots per cell in MS patients if DAPI was the definition of a cell (although I now see in M and M that it was total blue cells, not that a red signal had to necessarily be localized to a blue signal). Perhaps this can also be discussed, including the possibility that it might be this one individual in which this was the case.

 Response: thanks for the comment. While it is true that we have detected changes in the DAPI signal, these differences seem to be due to the different processing of the samples by the various brain blanks, which not only affected the DAPI signal but also the intensity of the PLA one. However, this did not affect the quantification of the number of heteromers per cell. Honestly, we think that the information that we can add to the manuscript would be merely speculative.

  In M and M, it is not clear what ab-crystallin stain defines.

 Response: thanks for raising this issue that has been considered in the revised version of the manuscript.

 Minor

  Line 60, “The” should be “They”; also is it true they participate in every event? Consider using “many” or “most”

 Response: we appreciate the suggestion that has been taken into account in the revised version of the manuscript.

Line 74, “is” should be “being”

 Response: thanks for the comment that has been considered in the revised version of the manuscript.

Line 121, “proved” should be “shown”

 Response: we appreciate the suggestion that has been taken into account in the revised version of the manuscript.

Line 122, “is” should be “was”; tenses are mixed throughout; consider changing all descriptions of results to past tense

 Response: thanks for the comment that has been considered in the revised version of the manuscript; all descriptions of results section have been changed to past tense.

Line 198, “consistenly” should be “consistent”

 Response: we appreciate the suggestion that has been taken into account in the revised version of the manuscript.

Observation: major changes are highlighted in yellow. Grammatical corrections are not highlighted.

Round 2

Reviewer 1 Report

Comments and Suggestions for Authors

The authors tried to address the comments or at least write about the limitations of the study